# Using a One Health approach to prioritize zoonotic diseases in China, 2019

Xin Wang[1], Jeanette J. Rainey[2], Grace W. Goryoka[3], Zuoru Liang[4], Shuyu Wu[2], Liming Wen[5], Ran Duan[1], Shuai Qin[1], Haodi Huang[6], Grishma Kharod[3], Carol Y. Rao[7], Stephanie J. Salyer[7], Casey Barton Behravesh[3], Huaiqi Jing[1]*

1 National Institute for Communicable Disease Control and Prevention, Chinese Center for Disease Control and Prevention, Beijing, China, 2 Division of Global Health Protection, United States Centers for Disease Control and Prevention, Beijing, China, 3 National Center for Emerging and Zoonotic Infectious Diseases, Centers for Disease Control and Prevention, Atlanta, GA, United States of America, 4 Center for Global Public Health, Chinese Center for Disease Control and Prevention, Beijing, China, 5 Yinchuan Animal Center for Disease Control and Prevention, Yinchuan, Ningxia, China, 6 Jiangsu Provincial Center for Disease Control and Prevention, Nanjing, Jiangsu, China, 7 Division of Global Health Protection, Centers for Disease Control and Prevention, Atlanta, GA, United States of America

* jinghuaiqi@icdc.cn

**Data Availability Statement:** All relevant data are within the manuscript.

**Funding:** The One Health Zoonotic Disease Prioritization Workshop was funded by the U.S.-China Collaborative Program on Emerging and Re-

## Abstract

### Background

China is vulnerable to zoonotic disease transmission due to a large agricultural work force, sizable domestic livestock population, and a highly biodiverse ecology. To better address this threat, representatives from the human, animal, and environmental health sectors in China held a One Health Zoonotic Disease Prioritization (OHZDP) workshop in May 2019 to develop a list of priority zoonotic diseases for multisectoral, One Health collaboration.

### Methods

Representatives used the OHZDP Process, developed by the US Centers for Disease Control and Prevention (US CDC), to prioritize zoonotic diseases for China. Representatives defined the criteria used for prioritization and determined questions and weights for each individual criterion. A review of English and Chinese literature was conducted prior to the workshop to collect disease specific information on prevalence, morbidity, mortality, and Disability-Adjusted Life Years (DALYs) from China and the Western Pacific Region for zoonotic diseases considered for prioritization.

### Results

Thirty zoonotic diseases were evaluated for prioritization. Criteria selected included: 1) disease hazard/severity (case fatality rate) in humans, 2) epidemic scale and intensity (in humans and animals) in China, 3) economic impact, 4) prevention and control, and 5) social impact. Disease specific information was obtained from 792 articles (637 in English and 155 in Chinese) and subject matter experts for the prioritization process. Following discussion of the OHZDP Tool output among disease experts, five priority zoonotic diseases were identified for China: avian influenza, echinococcosis, rabies, plague, and brucellosis.

emerging Infectious Diseases Cooperative Agreement #5U2GGH000961-04 (received by XW) and the National Sci-Tech Key Project 2018ZX10713003-002(received by HJ) and 2018ZX10713001-002 (received by XW).

**Competing interests:** The authors have declared that no competing interests exist.

## Conclusion

Representatives agreed on a list of five priority zoonotic diseases that can serve as a foundation to strengthen One Health collaboration for disease prevention and control in China; this list was developed prior to the emergence of SARS-CoV-2 and the COVID-19 pandemic. Next steps focused on establishing a multisectoral, One Health coordination mechanism, improving multisectoral linkages in laboratory testing and surveillance platforms, creating multisectoral preparedness and response plans, and increasing workforce capacity.

## Introduction

Zoonotic diseases are diseases that can spread between animals and people. Most known human infectious diseases and about three-quarters of newly emerging infections originate from animals. [1, 2] China is vulnerable to zoonotic disease transmission due to a large agricultural workforce, sizable domestic livestock population, and a highly biodiverse ecology, with more than 7,500 known vertebrate species [3]. The recent coronavirus disease 2019 (COVID-19) pandemic, resulting from the introduction of a novel coronavirus (SARS-CoV-2) to the human population, exemplifies challenges with emerging zoonotic diseases [4, 5]. Similarly, SARS-CoV, the virus that caused severe acute respiratory syndrome (SARS) in humans and spread worldwide in 2002–2003, is thought to have originated in bats before spreading to civets in a wildlife market in Guangdong Province, China [6]. COVID-19, SARS, and previous outbreaks of avian influenza highlight the potential global threat of zoonotic disease transmission and the need to use a multisectoral, One Health approach to address zoonotic diseases [7–9].

The population in China was approximately 1.4 billion in 2019 [10]. Most of this population resides in the temperate-monsoon and sub-tropical monsoon zones in Eastern China. The socio-economic level of development is also highest in eastern China. Western China, which borders Kazakhstan and Mongolia, is generally arid or semi-arid, less populated, relies on livestock production, and has a lower level of development. The tropical monsoon zone in the southern most regions of China experience zoonotic disease transmission risk similar to South East Asian countries, such as Laos, Myanmar and Vietnam, which border China to the south. Zoonotic disease transmission can impact society in many ways. This includes reducing the productivity of animals, threatening the livelihood of the population dependent on livestock as a major source of income, and causing unnecessary human illness and death [11].

Human and animal health and wildlife programs are implemented across 34 provinces, counties and prefectures, and autonomous regions under the authority and guidance of national level agencies. Chinese Centers for Disease Control (China CDC) and Animal Centers for Disease Control and Prevention (ACDC) conduct surveillance and investigations of zoonotic disease transmission among the human and livestock populations, respectively. The State Administration of Grasslands and Forestry and the Chinese Academy of Sciences monitor diseases among wildlife species, including occurrence of avian influenza in the wild bird population. To address the threat of zoonotic diseases in China, a One Health Zoonotic Disease Prioritization (OHZDP) workshop was held from in Beijing from May 15–17, 2019 [12, 13]. The goal of the workshop was to use a multisectoral, One Health approach to prioritize endemic and emerging zoonotic diseases of greatest concern in China that should be jointly addressed by sectors responsible for human, animal, and environmental health. The OHZDP workshop was supported by Chinese Center for Disease Control and Prevention (China CDC)

and United States Centers for Disease Control and Prevention (US CDC). Representatives from the United Nations Food and Agriculture Organization (FAO) and the World Health Organization (WHO) also participated in the workshop. Outcomes from the workshop can be applied to all emerging and re-emerging zoonotic diseases in China.

## Methods

### Workshop representatives

A total of 53 representatives from national and eight provincial-level human, animal, and environmental health sectors in China and international organizations participated in the workshop. The eight provinces represent the primary ecological and socio-demographic regions in the east (Jiangsu and Shandong), northwest (Qinghai, Ningxia, and Gansu) and southwest (Guangxi, Sichuan and Yunnan) of China respectively. Not all provinces were able to participate. However, national managers participating in the workshop were able to provide input on behalf of the human, animal, and environmental health sector staff in the provinces unable to attend. The 53 participants included 28 voting members, nine facilitators, and 16 advisors. Twenty-two of the 28 voting members were from provincial human, animal, and environmental health agencies, and the remaining six voting members were from national agencies. All voting members were managers for their agency or for the infectious disease unit in the agency (**Table 1**). To support future OHZDP workshops at the provincial and regional levels, three US CDC staff trained six local partners from the human and animal health sectors to serve as workshop facilitators. Sixteen advisors from international organizations and China CDC were present to observe and participate in discussions.

### Zoonotic disease list and literature review

The OHZDP process relies on a mixed methods prioritization approach based on a quantitative scoring tool developed by US CDC. The process has been previously described in detail [12, 13]. In brief, the OHZDP process allows participants to customize the ranking criteria and questions and weights to reflect the country-specific zoonotic disease situation and prevention and control goals across multiple sectors. In the three months before the OHZDP workshop, human and animal health staff in China developed an initial list of zoonotic diseases to be considered as potential zoonotic diseases of greatest concern to China for prioritization. The initial zoonotic disease list was developed from human and animal reportable disease lists from China and also incorporated expert opinion. Of the 40 diseases included on the national notifiable human disease list, 25 were zoonotic infections. Of the 157 diseases included on the national notifiable animal disease list, 17 could infect humans and were not notifiable human diseases. Combining these resulted in a list of 42 zoonotic diseases. This combined list of 42 diseases was shared with national human, animal, and environmental health experts for review and consideration. The experts could include additional zoonotic diseases to the list and remove diseases that were unlikely to pose potential threats in China. The final list included 30 zoonotic diseases for prioritization. The goal of the workshop was to identify zoonotic diseases of greatest concern to the country, and of those, prioritize the top five. For consistency with publications resulting from previous workshops, we use the term zoonotic diseases rather than zoonotic infections. The term highlights that these priority zoonotic infections are capable of causing disease.

We reviewed animal and human case reports and English and Chinese published and grey literature from the last five to 10 years to understand the occurrence and risks for each of the 30 diseases. Fourteen staff searched English and Chinese literature published to collect information on the prevalence, morbidity, mortality, and Disability-Adjusted Life Years (DALYs)

**Table 1. Agencies participating in the one health zoonotic disease prioritization workshop, Beijing, China, May 15–17, 2019.**

| Voting Members (No. participants) |
|---|
| *National Level* |
| National Institute for Communicable Diseases Control and Prevention, China CDC (2) |
| National Institute of Parasitic Diseases (NIPD), China CDC (1) |
| Division of Infectious Disease, China CDC (1) |
| China Animal Disease Control Centre (1) |
| National Research Center for Wildlife Diseases (1) |
| *Eastern Region* |
| Jiangsu Provincial CDC (2) |
| Shandong Provincial CDC (1) |
| Jiangsu Academy of Agricultural Sciences, Jiangsu Society of Animal Husbandry and Veterinary Medicine (1) |
| Shandong Provincial Animal CDC (1) |
| *Northwestern Region* |
| Qinghai Provincial CDC (3) |
| Gansu Provincial CDC (1) |
| Ningxia Hui Autonomous Region CDC (1) |
| Qinghai Animal CDC (2) |
| Lanzhou Veterinary Research Institute, Chinese Academy of Agricultural Sciences (1) |
| Qinghai Academy of Animal Science and Veterinary Medicine (3) |
| *Southwestern Region* |
| Yunnan Provincial CDC (1) |
| Guangxi University of Chinese Medicine, School of Public Health and Management (1) |
| Sichuan Provincial CDC (1) |
| Guangxi Zhuang Autonomous Region Animal CDC (1) |
| Yunnan Provincial Institute of Endemic Disease Prevention and Control (1) |
| Sichuan Provincial Animal CDC (1) |
| **Facilitators** |
| US Centers for Disease Control and Prevention (Trainer) |
| National Institute for Communicable Diseases Control and Prevention, China CDC |
| Center for Global Public Health, China CDC |
| Jiangsu Provincial CDC |
| National Research Center for Wildlife Diseases |
| Yinchuan Animal CDC |
| **Advisors and Other Participants** |
| US Department of Health and Human Services China Office |
| US Department of Agriculture (USDA) China Office |
| US National Institutes of Health |
| US Centers for Disease Control and Prevention |
| Food and Agriculture Organization (FAO) China Office |
| WHO China Office |
| Center for Global Public Health, China CDC |
| National Institute for Communicable Diseases Control and Prevention, China CDC |

for each of the 30 diseases from China and the WHO Western Pacific Region. Staff identified relevant articles by using Boolean logic and search terms that included country name (China), disease name (e.g., brucellosis), and one of the following terms: "morbidity", "mortality", "DALYs", "cases", "animals", "vaccine", and "wildlife". If regional data were not available,

global estimates on prevalence, incidence, morbidity, mortality, and DALYs were used instead. Regional and global data were obtained from websites of the WHO and FAO. All corresponding data were saved and shared with workshop participants for future reference.

### Criteria selection

During the workshop, voting members jointly identified five criteria for ranking the 30 zoonotic diseases. Representatives also selected a single categorical question for each criterion. All questions required an ordinal or multinomial answer to be entered into the OHZDP Tool. Once the criteria and questions were identified, the nine voting groups separately decided on the relative importance of each criterion. Each group's ranking was then entered in the OHZDP Tool by a facilitator and a group weight for each criterion was calculated to reflect this relative importance.

### Disease weighting and final ranking

Facilitators and representatives answered each question for the 30 zoonotic diseases using country-specific, regional, and global data collected previously through the literature review. After scoring all the zoonotic diseases, the OHZDP Tool uses a decision tree analysis to generate the zoonotic disease ranking [12, 13]. Each criterion's weight is multiplied by the score assigned for each disease and is weighted by the number of answer choices. The scores for all five questions for each disease were summed and then normalized such that the highest final score was 1. The ranked list of zoonotic diseases and corresponding normalized scores were presented to the representatives to discuss and decide on a final list of priority zoonotic diseases for China.

## Results

The final list of 30 zoonotic diseases discussed at the One Health workshop in China included diseases caused by bacteria (n = 14), viruses (n = 9), parasites (n = 6), and a prion pathogen (n = 1) (**Table 2**). A total of 792 published articles were identified and reviewed for information on these 30 zoonotic diseases in China and the surrounding region. This included 637 and 155 articles from English and Chinese journals, respectively. Data from these articles, in addition to information obtained directly from WHO and FAO websites, were used to address the ordinal or multinomial question developed for each ranking criterion. These criteria and questions are described in **Table 3**.

When the workshop participants reviewed the initial list of 29 zoonotic diseases for prioritization, they discussed and decided to also include Japanese Encephalitis, making the final zoonotic disease list for prioritization 30 diseases. A rapid review of the literature and relevant data sources was conducted for Japanese encephalitis, a vector-transmitted virus. A list of zoonotic diseases and their normalized scores was generated from the OHZDP Tool (**Table 2**). The 28 voting members discussed the disease ranking and finalized a priority zoonotic disease list for China. Although the normalized final score for brucellosis was 0.690 (rank 8), the disease was added to the final list of the top five priority zoonotic diseases based on discussions among the voting members. The final priority zoonotic disease list for China included avian influenza, echinococcosis, rabies, plague, and brucellosis. Epidemiologic data for each of these diseases is included in the workshop report [13].

After finalizing the list of priority zoonotic diseases, workshop representatives were divided into four groups to discuss next steps, plans, and approaches to effectively address the top five priority zoonotic diseases for China. Topics discussed included strengthening One Health

**Table 2. Raw and normalized scores for 30 zoonotic diseases generated from the One Health Zoonotic Disease Prioritization (OHZDP) tool during the OHZDP workshop, Beijing, China, May 15–17, 2019.** The top five priority zoonotic diseases selected by voting members are highlighted in grey and bolded.

| Rank | Disease | Raw Score | Normalized Final Score |
|---|---|---|---|
| **1** | **Zoonotic avian influenza** | **0.975** | **1.000** |
| **2** | **Echinococcosis/hydatid diseases** | **0.927** | **0.951** |
| **3** | **Rabies** | **0.925** | **0.949** |
| **4** | **Plague** | **0.777** | **0.797** |
| 5 | Zoonotic tuberculosis | 0.754 | 0.773 |
| 6 | Streptococcus suis | 0.720 | 0.739 |
| 7 | Japanese Encephalitis | 0.677 | 0.694 |
| **8** | **Brucellosis** | **0.672** | **0.690** |
| 9 | Schistosomiasis | 0.667 | 0.684 |
| 10 | Scrub typhus | 0.664 | 0.681 |
| 11 | Creutzfeldt-Jakob disease | 0.651 | 0.667 |
| 12 | Anthrax | 0.597 | 0.613 |
| 13 | Severe fever with thrombocytopenia syndrome (SFTS) | 0.588 | 0.603 |
| 14 | Cysticercosis | 0.578 | 0.593 |
| 15 | Hepatitis E | 0.556 | 0.570 |
| 16 | Q fever | 0.508 | 0.521 |
| 17 | Salmonellosis | 0.476 | 0.488 |
| 18 | Cryptosporidiosis | 0.476 | 0.488 |
| 19 | Colibacillosis | 0.454 | 0.465 |
| 20 | Listeriosis | 0.431 | 0.442 |
| 21 | Severe Acute Respiratory Syndrome (SARS) | 0.397 | 0.407 |
| 22 | Tularemia | 0.378 | 0.387 |
| 23 | Xinjiang hemorrhagic fever (XHF) Crimean Congo hemorrhagic fever (CCHF) | 0.378 | 0.387 |
| 24 | Clonorchiasis | 0.377 | 0.386 |
| 25 | Yersiniosis | 0.353 | 0.362 |
| 26 | Leptospirosis | 0.323 | 0.331 |
| 27 | Intestinal amebiasis | 0.308 | 0.316 |
| 28 | Epidemic hemorrhagic fever | 0.307 | 0.315 |
| 29 | Zoonotic swine influenza | 0.246 | 0.252 |
| 30 | Campylobacteriosis | 0.198 | 0.203 |

coordination, surveillance and laboratory, preparedness and response, and workforce capacity. Recommendations were aggregated and presented to all representatives (**Table 4**).

## Discussion

The OHZDP workshop for China held in May 2019 provided a platform for establishing a One Health approach to emerging and re-emerging zoonotic diseases. Although the China disease experts developed country-specific ranking criteria and questions, the priority disease list (avian influenza, echinococcosis, rabies, plague, and brucellosis) and recommended next steps were consistent with outcomes from workshops conducted in other countries [13–18]. Coronaviruses were not included in the list of the top five priority zoonotic diseases in China at the time of the workshop. However, the COVID-19 pandemic has further demonstrated the importance of a One Health approach to emergency preparedness and response [19, 20]. Human, animal, and environmental health sectors have collaborated in China to investigate and respond to COVID-19 [21], SARS, and avian influenza disease transmission through information sharing, environmental cleaning, and case detection and management [22–24].

**Table 3. Criteria, criterion weights, questions, and answer choices selected by workshop participants, Beijing, China, May 15–17, 2019.**

| Criteria | Criterion Weight | Question | Answer Choices |
|---|---|---|---|
| Disease hazard/ severity | .403 | What is the case fatality rate (CFR) in humans? | i. ≤ 1% CFR (0)<br>ii. 1% < x ≤ 5% CFR (1)<br>iii. 5% < x ≤ 10% CFR (2)<br>iv. 10% < x ≤ 50% CFR (3)<br>v. > 50% CFR (4) |
| Epidemic scale and intensity | .277 | In the past 5 years, has the disease caused outbreaks or sporadic cases in China? | i. Yes–both animals and humans (3)<br>ii. Yes–no animals, just humans (2)<br>iii. Yes–just animals, no humans (1)<br>iv. No–neither animals nor humans (0) |
| Economic impact | .123 | What is the livestock production loss*? | i. Less than or equal to 5% production loss (0)<br>ii. > 5% to 10% production loss (1)<br>iii. > 10% to 20% production loss (2)<br>iv. >20% production loss (3) |
| Prevention & control | .099 | Are prevention and control measures readily available? | i. None (4)<br>ii. Human vaccine, or human treatment/medicine, or animal vaccine (3)<br>iii. Human vaccine, and human treatment/medicine, no animal vaccine (2)<br>iv. Human vaccine, and animal vaccine, no human treatment/medicine (1)<br>v. Human vaccine, and human treatment/medicine, and animal vaccine (0) |
| Social impact | .095 | How much public attention has the disease received in the past 1 year according to the Baidu index[†]? | i. < 100 hits[+] (0)<br>ii. 101–300 hits[+] (1)<br>iii. 301–500 hits[+] (2)<br>iv. 501–700 hits[+] (3)<br>v. > 700 hits[+] (4) |

*Case fatality rate and mortality were used as proxies in cases where livestock production loss numbers were not available; global data were used due to limited China-specific data. For diseases where no data were available, a score of 0 was applied.

[†]http://index.baidu.com/v2/index.html#/ The Baidu index represents the weighted sum of the search frequency in a certain time period. Higher index values reflect a larger search volume, suggesting greater public interest and social impact

[+] A hit is a request to a web server for a file (e.g. image, pdf, etc.) or an HTML page that contains specific content being searched.

Workshop representatives supported efforts to strengthen multisectoral, One Health collaboration for preparing and responding to outbreaks caused by other emerging and re-emerging zoonotic diseases.

In China, One Health preparedness and planning collaboration could focus on identifying and assigning roles and responsibilities of staff from each sector as well as establishing mechanisms for human and animal case finding, standard laboratory testing and further information sharing, all in advance of an emergency. National and provincial level-led One Health offices could coordinate these activities to provide appropriate authority and resources as well as assist in conducting joint outbreak investigations, developing joint risk communication strategies to provide up-to-date and consistent messaging [24] and ensuring a stockpile of available vaccines and treatments for both humans and animals are available, if and when needed [25, 26].

Representatives also recommended regular multisectoral, One Health trainings, simulation exercises, and other preparedness and outbreak response educational opportunities to help strengthen the One Health workforce in China for responding to future zoonotic disease emergencies. Engagement of stakeholders across agencies in planning and preparedness activities can facilitate One Health collaboration, coordination, and communication before and during public health emergencies [26, 27]. China CDC has previously conducted pandemic influenza

**Table 4. Recommendations and next steps identified by workshop participants for improving the control and prevention of the top five priority zoonotic diseases in China.** The priority zoonotic diseases for China include avian influenza, echinococcus, rabies, plague, and brucellosis.

| Recommendation | Proposed Activities |
|---|---|
| Strengthen One Health Coordination | Develop framework (above Health Commission and Department of Agriculture) to coordinate multisectoral work on priority zoonotic diseases. |
| | Develop and implement multisectoral plans for disease control and prevention (including for outbreak control and response). |
| | Establish platform for multisectoral information sharing |
| Support Multisectoral, One Health Surveillance and Laboratory System Development | Identify and standardize common data elements/variables to facilitate multisectoral data sharing. |
| | Develop systems/platforms for routine multisectoral data sharing. |
| | Establish collaborative multisectoral process for sharing and discussing lessons learned on priority zoonotic disease prevention and control strategies. |
| | Identify resources to support training/capacity building on One Health surveillance approaches and laboratory testing. |
| Collaborate in outbreak preparedness and response | Ensure a joint stockpile of drugs, vaccine, personal protective equipment (PPE), and laboratory supplies are available to all sectors for the priority zoonotic diseases. This potential cost-savings approach can help strengthen and standardize response capacity. |
| | Develop joint One Health trainings, simulation exercises, and other laboratory and response staff across sectors for outbreak/pandemic preparedness for the priority zoonotic diseases. |
| | Develop and pilot mechanism to rapidly share outbreak related information and data across relevant sectors to improve response capacity for priority zoonotic diseases. |
| | Create and disseminate One Health education material and information sharing sessions to general public on potential zoonotic disease threats. |
| Strengthen Workforce Capacity | Develop One Health training opportunities during pre-service medical and veterinary school programs, including training on One Health concepts to encourage future professional collaboration. |
| | Include information on biosecurity and biosafety concepts on the five priority zoonotic diseases in pre-service and in-service programs. |
| | Organize in-service One Health seminars, presentations, and trainings on the five priority zoonotic diseases, including exposure risks to human and animal health workers. |

exercises. Expanding such exercises to include other sectors and address priority zoonotic diseases would likely be helpful.

All five of the top priority zoonotic diseases for China identified during the workshop have been included in the National Notifiable Disease Reporting System (NNDRS) maintained by National Health Commission and the China CDC for human infections [28]. China CDC also maintains surveillance for certain animal infections such as the plague. Human infections resulting from these five priority zoonotic diseases are typically identified, investigated, and reported to NNDRS by local and provincial level Centers for Disease Control (CDCs). Diseases in animals and wildlife are investigated through local level agencies and reported to separate national surveillance systems [29]. Laboratory diagnostic capacity and testing procedures vary by disease and sector, which can limit control and prevention efforts. Human and dog rabies,

ranked third on the priority zoonotic disease list, are notifiable to human and animal health agencies in China. However, only 2% of all human cases of rabies are laboratory confirmed and the number of reported dog rabies was far below the number of human cases in 2017 (75 dog cases compared to 502 human cases) [29, 30]. During a Stepwise Approach to Rabies Elimination assessment in China in March 2019, human and animal health sector staff advocated for development of systems to jointly collect, view, and analyze human and animal data to inform provincial and national-level decision making related to rabies elimination program activities [31]. Similar One Health approaches would likely be beneficial for preventing and controlling the other priority zoonotic diseases for China.

OHZDP workshop participants from other countries made similar recommendations for developing systems or mechanisms to organize and harmonize data collection (including standardization) across the relevant different sectors for the priority zoonoses [32]. Such systems could be implemented at the national and provincial level in China and include feedback tools (e.g., dashboards for visualization) for key decision makers in each sector [33, 34]. Animal disease surveillance, including at animal farms, breeding sites, and market locations are critical for early detection of zoonotic diseases and limiting transmission or 'spill-over' to the human population. Animal disease surveillance could be highly beneficial for all five priority zoonotic diseases, including brucellosis where the transmission between animal and human cases may be through direct exposures, or indirect, through consumption of unpasteurized milk. Pilot programs using mobile phones and tablets to collect and transmit information for geographically disparate locations to monitor and visualize disease and risk factor data in close to real-time for both human and animal populations can improve the efficiency of identifying and responding to zoonotic disease transmission [35–37].

Several provinces have implemented multi-sector trainings prior to the OHZDP workshop. Provinces along the Tibetan Plateau in western China (e.g., Qinghai, Tibet, and Xinjiang), for example, have conducted multisectoral annual meetings and trainings on echinococcosis prevention and control due to the endemicity of the infection among herding dogs and potential for debilitating health effects for the human population [38, 39]. Other provinces (e.g., Inner Mongolia) have provided targeted multisectoral, One Health training through the National Field Epidemiology Training Program for specific diseases, such as brucellosis (unpublished data). However, these multisectoral training opportunities are limited and do not yet exist in all provinces or for all priority zoonotic diseases. OHZDP workshop representatives in China supported the development of additional opportunities during pre-service medical and veterinary school programs that can encourage future professional collaboration through the One Health approach to zoonotic disease control and prevention. National and provincial level programs can also expand knowledge of biosecurity and biosafety concepts for priority zoonotic diseases during pre-service and in-service programs and multisectoral training through implementation of professional development projects that require engagement of staff from other health sectors. These One Health training programs can support future work for all emerging and re-emerging infectious diseases.

The OHZDP process is flexible and adaptable to address zoonotic diseases at subnational, national, and regional levels. The main outcome of this workshop was to identify a list of priority zoonotic diseases for China that could be addressed using a multisectoral, One Health approach. The priority zoonotic disease list was developed prior to the emergence of SARS-CoV-2 and the COVID-19 pandemic. We hypothesize that COVID-19 would likely score among the top five zoonotic diseases if we conducted this workshop during 2020, primarily due to significant transmission of SARS-CoV-2 and COVID-19 disease. We argue that regardless of individual disease scores, the next steps outlined by representatives from this workshop can be applied to COVID-19. Proposed work on surveillance, multisector training, and

preparedness and response capacity can be applied to any emerging or re-emerging zoonotic disease. Future One Health workshops can be conducted to review and update the priority zoonotic disease list by including assessments of SARS-COV2 and other coronaviruses, as warranted. Such workshops can provide additional opportunities to further strengthen and institutionalize multisectoral, One Health collaboration on human, animal, and environmental health priorities.

The OHZDP workshop has been conducted in more than 25 locations [13]. Each of the countries, including China, developed country-specific criteria and corresponding weights for ranking the zoonotic diseases of greatest national concern. In China, these country-specific criteria included the use of a 'social impact' criterion to reflect public interest and concern based on Baidu (internet) search analytic results. The one-year Baidu search timeframe could have influenced the score of this criterion. The index score of certain diseases could have reflected possible mentions from other news sources and/or international events as well as zoonotic diseases present during the year proceeding the workshop. However, the weight for this criterion was relatively lower compared to the other four criteria. Using 'social impact' to reflect public interest and concern was unique and an informative criterion that incorporated public perspectives in ranking the relative importance of zoonotic diseases in China. Utilizing a prioritization method that can be adapted to meet country level concerns promoted engagement and contribution from workshop participants [12, 13]. Representatives in China also decided to score zoonotic diseases without preventable measures (such as treatment or a vaccine) higher compared to vaccine preventable diseases or those with a known or accessible treatment. This differed from other workshops [13–18] and was based on the perceived need to rapidly respond to zoonotic diseases having the potential for uncontrolled epidemic spread. Despite having different criteria and questions, two zoonotic diseases, rabies and avian influenza, were among the top five prioritized zoonotic diseases in 25 and 24 of the workshops conducted to date, respectively. The national level workshop did not aim to address all possible zoonotic disease prioritization criteria since the OHZDP methods focus on the top five criteria of greatest concern. Participants discussed many different criteria during the workshop and agreed on the five criteria most likely associated the potential zoonotic disease threat in the country. Future workshops–including those conducted at the regional or provincial level—could develop and apply more targeted and specific criteria for identifying a local list of priority zoonotic diseases. The outcomes of this first OHZDP workshop in China support opportunities for international engagement, regional collaborations, and sharing of lessons learned to develop and implement multisectoral, One Health programs to prevent and control these prioritized zoonotic diseases.

## Conclusion

Representatives from Chinese national and provincial level human, animal and environment health sectors participated in an OHZDP workshop in May 2019. Workshop representatives successfully developed a list of priority zoonotic diseases that can be jointly addressed through a multisectoral, One Health approach. Voting members agreed upon criteria and questions specific to China for ranking 30 zoonotic diseases. Combining results from the OHZDP Tool and subject matter expert discussions, representatives agreed on five priority zoonotic diseases for China: avian influenza, echinococcus, rabies, plague, and brucellosis. The OHZDP workshop provided an opportunity for multisectoral, One Health discussion and collaboration. As a product of this collaboration, representatives suggested the establishment of a One Health coordinating office to help facilitate information sharing among One Health sectors. Approaches for improving One Health surveillance and laboratory capacity for the priority

zoonotic diseases, particularly in responding to outbreaks were also identified. To respond to the priority zoonotic diseases, opportunities to strengthen One Health workforce capacity should be considered at the local and provincial levels. One Health collaboration in these areas and others can help reduce unnecessary morbidity and mortality and potential economic burden due to emerging and re-emerging zoonotic diseases in China. Future workshops can be conducted to review and update the list of priority zoonotic diseases and further strengthen multisectoral, One Health collaboration in China.

## Acknowledgments

We thank the representatives who attended and participated in the China One Health Zoonotic Disease Prioritization Workshop. Their expertise and contribution were vital to the success of the workshop.

The findings and conclusions in this report are those of the author(s) and do not necessarily represent the official position of the Chinese Center for Disease Control and Prevention nor the United States Centers for Disease Control and Prevention.

## Author Contributions

**Conceptualization:** Xin Wang, Jeanette J. Rainey, Grishma Kharod, Carol Y. Rao, Stephanie J. Salyer, Huaiqi Jing.

**Formal analysis:** Xin Wang, Jeanette J. Rainey, Grace W. Goryoka, Zuoru Liang, Shuyu Wu, Liming Wen, Ran Duan, Shuai Qin, Haodi Huang, Grishma Kharod, Carol Y. Rao, Stephanie J. Salyer, Casey Barton Behravesh.

**Funding acquisition:** Xin Wang, Huaiqi Jing.

**Investigation:** Xin Wang, Jeanette J. Rainey, Grace W. Goryoka, Zuoru Liang, Shuyu Wu, Liming Wen, Ran Duan, Shuai Qin, Haodi Huang, Grishma Kharod, Carol Y. Rao, Stephanie J. Salyer.

**Project administration:** Xin Wang, Jeanette J. Rainey, Huaiqi Jing.

**Writing – original draft:** Xin Wang, Jeanette J. Rainey, Grace W. Goryoka, Zuoru Liang, Casey Barton Behravesh.

**Writing – review & editing:** Xin Wang, Jeanette J. Rainey, Grace W. Goryoka, Zuoru Liang, Shuyu Wu, Liming Wen, Ran Duan, Shuai Qin, Haodi Huang, Grishma Kharod, Carol Y. Rao, Stephanie J. Salyer, Casey Barton Behravesh, Huaiqi Jing.

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
