## [Decision Letter · Decision Letter 0]

25 Feb 2021

PONE-D-20-41070

Using a One Health Approach to Prioritize Zoonotic Diseases in China, 2019

PLOS ONE

Dear Dr. Jing,

Thank you for submitting your manuscript to PLOS ONE. After careful consideration, we feel that it has merit but does not fully meet PLOS ONE’s publication criteria as it currently stands. Therefore, we invite you to submit a revised version of the manuscript that addresses the points raised during the review process.

All the three reviewers recommend that you make minor revisions to your manuscript. Please attend to all the concerns raised by the reviewers and submit a revised manuscript.

We look forward to receiving your revised manuscript.

Kind regards,

Martin Chtolongo Simuunza, PhD

Academic Editor

PLOS ONE

Journal Requirements:

2. We note that Figure 1 in your submission contains map images which may be copyrighted. All PLOS content is published under the Creative Commons Attribution License (CC BY 4.0), which means that the manuscript, images, and Supporting Information files will be freely available online, and any third party is permitted to access, download, copy, distribute, and use these materials in any way, even commercially, with proper attribution. For these reasons, we cannot publish previously copyrighted maps or satellite images created using proprietary data, such as Google software (Google Maps, Street View, and Earth). For more information, see our copyright guidelines: http://journals.plos.org/plosone/s/licenses-and-copyright.

(1) You may seek permission from the original copyright holder of Figure 1 to publish the content specifically under the CC BY 4.0 license. 

Reviewers' comments:

Reviewer's Responses to Questions

**Comments to the Author**

1. Is the manuscript technically sound, and do the data support the conclusions?

Reviewer #1: Partly

Reviewer #2: Yes

Reviewer #3: Yes

2. Has the statistical analysis been performed appropriately and rigorously? 

Reviewer #1: N/A

Reviewer #2: Yes

Reviewer #3: N/A

3. Have the authors made all data underlying the findings in their manuscript fully available?

Reviewer #1: No

Reviewer #2: Yes

Reviewer #3: No

4. Is the manuscript presented in an intelligible fashion and written in standard English?

Reviewer #1: Yes

Reviewer #2: Yes

Reviewer #3: Yes

5. Review Comments to the Author

Reviewer #1: “Using a One Health Approach to Prioritize Zoonotic Diseases in China, 2019”

PLOS ONE

This article describes findings from a May 2019 One Health Zoonotic Disease Prioritization (OHZDP) workshop in China. Representatives were identified from eight provinces, national agencies, and international agencies, and voting members selected from thirty zoonotic diseases to create a zoonotic disease prioritization list consisting of five diseases: avian influenza, echinococcosis, rabies, plague, and brucellosis. The authors emphasize a One Health, multi-sectoral approach throughout the document. The importance of the workshop and its applicability for zoonotic disease preparedness is evident.

The workshop organization, process of disease evaluation and prioritization, and recommendations were poorly outlined. The workshop’s country-specific use for China was described, but as described by the authors had a bias towards the richer more populous southeastern region and had very limited representation from the West, Northeast or central regions. Despite this, echinococcosis made the final list, an issue that was addressed in a zoonoses workshop held in the Western provinces. How was information from other regional provinces weighed into this process? This is largely unclear.

Overall, this article was written with broad brush strokes and lacks the granularity needed to understand the actual process, biases, concerns of methods and results.

• Methods: How were the 53 representatives chosen? How were the 8 provinces chosen to be represented? How were the 29 voting members chosen?

• Methods: The authors explain how 30 zoonotic diseases were researched for finding DALYs, morbidity, mortality, and prevalence; however, it is unclear how the 30 zoonotic diseases were originally picked. Lines 136-137 could be more specific explaining the initial zoonotic disease list development. Why wasn’t a more general literature review used that might help catch emerging infections that are not reportable but could be greater problems in the future?

• Throughout the paper the term “zoonotic diseases” is used although the actual transmissible agent(s) are zoonotic infections. More care should be taken to describe infection vs. disease. This includes presenting causative agents in table 2 so that the other 27 countries that have also gone through this process can determine areas of overlap based on the actual microbes vs. the generalized disease.

• Lines 176-177: There is a complete lack of explanation how the criteria were determined and very limited justification of them, despite the fact that this is described as a discussion. For instance why was human mortality the final determinant for disease hazard/burden vs. DALY or something that also included agricultural impact? Why was vaccination the only control/prevention measure listed in the literature review and as the determining criteria? What is the evidence behind internet hits (Baidu index) as a measure of “societal impact” particularly in a society that does not have free internet access to all sites and data? What are the inherent biases of these choices?

• Expand upon weights, and questions/answers within the text- how did the answers determined using the criteria in Table 3 lead to the raw scores in table 2? This must be explained so that these numbers make sense.

• Since the focus is on zoonotic diseases, it would be helpful to show the morbidity/mortality numbers for animals affected by the diseases chosen in addition to the human data.

Minor points:

• Update the population of China stated in line 84.

• Add international agencies that participated in the workshop among those listed in Table 1.

• Why is Xinjiang hemorrhagic fever highlighted in Table 2?

Reviewer #2: The authors report on the results of a One Health meeting to prioritize zoonotic disease control measures in China. The meeting was sponsored by Chinese agencies responsible for human, animal and environmental health, with assistance from the US CDC. The meeting was held in 2019, prior to the start of the COVID-19 pandemic. As such COVID was not subject to this prioritization effort. However, the authors comment that “outcomes from the workshop can be applied to all emerging and re-emerging zoonotic diseases in China. (line 181). They also note in the discussion that the COVID-19 pandemic has further demonstrated the importance of a One Health approach to emergency preparedness and response (lines 207-208). I think it would be useful for the authors to demonstrate how the scaling system could be used to compare SARS-CoV2 to the other zoonotic diseases they evaluated. By my reckoning it comes out ahead of avian influenza, although the balance of the assessment would shift from early in the pandemic (no vaccine or therapies, little awareness) to later in the pandemic (vaccine and therapies and huge public interest). This would take a fairly dry assessment of an important exercise and turn it into a model for how this approach can help reprioritize public health actions.

Specific comments:

Lines 47-48 “Disease severity” would be more understandable than “disease hazard” in describing this measure.

Line 76 “resulting from spillover… which might be from an animal source.” The concept of spillover implies an animal source. Why is it necessary to qualify the statement?

Line 109 As noted above, why not demonstrate how to apply the outcomes to the COVID-19 pandemic?

Line 120 Note at the time of the meeting.

181 What is the point of a 6-decimal place “normalized final score”? The ranking is based on categorical values from 0-4. The decimals imply a level of precision that are not reasonable distinctions. Particularly when the final choice was made based on “discussions among the voting members”.

Lines 194-197 This part of the discussion is a restatement of methods.

Lines 198-199 This is a restatement of results.

Lines 200-204 This appears to be a restatement of the premise behind the meeting.

Lines 315-317 This should be part of an acknowledgement and not part of the conclusions.

In looking at the scoring system, it is not clear why the prevention and control measures are expressed as a negative relation, when all of the others are positive. I can understand that if a disease emerges for which there is no vaccine or therapy available (see COVID-19) then prioritizing their development is important. However, if (such as with rabies) vaccines and therapies are available, and the disease is not being effectively controlled, that demonstrates a critical need to better coordinate measures across One Health platforms. For the diseases that were ranked these are important considerations.

Reviewer #3: Overall, this is a nicely done paper and illustrates the utility of systematic prioritization of known zoonotic disease threats.

Methods: In the methods section the workshop representatives are a bit confusing. I would suggest modifying the Table 1 to include columns where they can check off if it is human, animal or environment. And possibly to indicate if they had a voting member. The 9 groups were also unclear – how were they divided and why?

Also the totals don’t seem to add up - there were 53 participants but then there were 28 voting members but 16 observers. This is 56. Just a bit more clarity here would be very helpful as to the relative roles and who was involved in the decision making.

The lit review and data gathering process is critical. It is great that they did this. I would suggest that they include a bit more detail on the databases they searched, time frames searched, etc. While not necessarily a systematic review, it would be great to have as much detail reported about the search as possible. Also, were things like Promed databases used, etc.

There are nine groups with 28 voting members – but the groups had to come to consensus so does this really mean there are only 9 votes?

This is a very nice discussion and systematic method of prioritizing current identified threats. However, as was pointed out SARS-CoV-2 and coronaviruses were not included at the time of the workshop but emerged as a significant threat. Some discussion around how to increase the capacity and response for unknown threats would also be useful.

6. PLOS authors have the option to publish the peer review history of their article (what does this mean?). If published, this will include your full peer review and any attached files.

Reviewer #1: No

Reviewer #2: No

Reviewer #3: No

---

## [Author Response · Author response to Decision Letter 0]

12 Jul 2021

Responses to Editor and Reviewers

Editor’s comments

Thank you for this reminder. We have reviewed and verified that the revised manuscript meets PLOS ONE's style requirements, including those for file naming.

2. We note that Figure 1 in your submission contains map images which may be copyrighted. We require you to either (1) present written permission from the copyright holder to publish these figures specifically under the CC BY 4.0 license, or (2) remove the figures from your submission.

Thanks for your suggestion. We agree to remove Figure 1 from submission.

Reviewers' comments

Reviewer #1

1. The workshop’s country-specific use for China was described, but as described by the authors had a bias towards the richer more populous southeastern region and had very limited representation from the West, Northeast or central regions. Despite this, echinococcosis made the final list, an issue that was addressed in a zoonoses workshop held in the Western provinces. How was information from other regional provinces weighed into this process? This is largely unclear.

We appreciate the reviewer’s question. From our perspective, the selection of provinces may have biased slightly away from the populated and highly developed Eastern Provinces. We selected provinces to represent the epidemiology of zoonotic infections in each of the major regions in China. Animal and human health sector staff from Qinghai, Ningxia, Gansu, Guangxi and Yunnan Provinces where are in the western region of China participated in the workshop. Qinghai, Ningxia, and Gansu provinces have lower population density and socio-economic development compared to provinces in the Eastern Region, and experience zoonotic disease transmission similar to other provinces in the Northwest Region, including Inner Mongolia, Xinjiang, and Tibet. Qinghai Province, for example, is located on the northeastern part of the Tibetan plateau with an average elevation of 3,000 meters. Guangxi Yunnan Provinces both have significantly lower GDP compared to the populated Eastern Provinces.

Information about zoonotic disease transmission from provinces in the central (e.g., Hubei, Hunan) and northeastern (e.g., Jilin and Heilongjiang) region were obtained from review of the published and grey literature, official reports, and input from National Animal, Human, and Environmental sector staff in Beijing. For example, Yin Wenwu and his team from China CDC worked closely with Heilongjiang and Hunan Province, the high-risk province for rabies and anthrax respectively.

Although not all provinces were able to participate, the pre-workshop review and discussions during the workshop provided the opportunity to review existing animal, human, and environmental data and information from all regions in China. In future workshops, we will request participation from these other provinces.

2. Methods: How were the 53 representatives chosen? How were the 8 provinces chosen to be represented? How were the 29 voting members chosen?

The total of 53 participants included 28 voting members, 16 advisors, and 9 facilitators. Twenty-two of the voting members were from provincial human, animal, and environmental health agencies in the east, northwest and southwest of China and the remaining six voting members were from national agencies. All voting members were the managers responsible for their agency or for the infectious disease unit in the agency. Nine advisors were from international agencies in China and the other seven were from China CDC. Three of the 9 facilitators were from US CDC and served as trainers. The other six facilitators were from China CDC and other agencies in China. Agency representatives presented in Table 1. We have included this additional information to the Methods section of the revised manuscript. Please see lines 101-108. Table 1 was also revised to add agencies not included as voting members. 

3. Methods: The authors explain how 30 zoonotic diseases were researched for finding DALYs, morbidity, mortality, and prevalence; however, it is unclear how the 30 zoonotic diseases were originally picked. Lines 136-137 could be more specific explaining the initial zoonotic disease list development. 

We initiated the selection of the 30 important zoonotic infections with a review of China’s national list of notifiable human and animal diseases. We identified the zoonotic infections from the notifiable list of human diseases and cross-referenced this with the list of reportable animal diseases that could cause human infections. Of the 40 diseases included on the national notifiable human disease list, 25 were zoonotic infections. Of the 157 diseases included on the national notifiable animal disease list, 17 could infect humans and were not notifiable human diseases. Combining these resulted in a list of 42 zoonotic diseases. This combined list of 42 diseases was shared with human, animal, and environmental health experts at the national offices in Beijing. The experts could also include zoonotic diseases that could be a concern in China but were not notifiable disease in the country. The final list included 30 zoonotic diseases for prioritization. The goal of the workshop was to identify zoonotic diseases of greatest concern to the country, and of those, prioritize the top five. We reviewed animal and human case reports from the last 5 to 10 years, reviewed the English and Chinese literature to understand the occurrence and risks for each of these 30 diseases. Additional zoonotic diseases could be added to the list during the workshop if the participants determined that another disease should be considered. With this approach, Japanese encephalitis (JE) was added to the list of the diseases during the workshop. We conducted a rapid review of the literature on JE and provided this information to workshop participants for discussion. The workshop was not intended to provide an exhaustive list of every possible zoonotic infection in China or the WHO Western Pacific Region. We have provided additional information to paper to better describe the process as outlined above. Please see lines 119-129 (methods) and lines 166-169 (results).

4. Why wasn’t a more general literature review used that might help catch emerging infections that are not reportable but could be greater problems in the future?

We developed the initial list of important zoonotic infections in China using the lists of national notifiable diseases and expert opinion. After developing the initial list of important zoonotic diseases in China, human health, animal health, and environmental health ministries reviewed the list and had the opportunity to add other zoonotic infections to the list is warranted (by expert opinion and/or other unpublished or published literature). During the workshop, additional zoonotic infections could be added to the list if requested by the participants. This is the standard approach used in the One Health Zoonotic Disease Prioritization workshop and has been determined to the be an efficient approach for the host countries. Because we also included feedback from experts on the list of important zoonotic diseases, we do not envision a larger literature search would have resulted in a different list of priority zoonotic diseases. 

5. Throughout the paper the term “zoonotic diseases” is used although the actual transmissible agent(s) are zoonotic infections. More care should be taken to describe infection vs. disease. This includes presenting causative agents in table 2 so that the other 27 countries that have also gone through this process can determine areas of overlap based on the actual microbes vs. the generalized disease.

Thank you for comment. We appreciate the distinction. As with the other One Health Zoonotic Disease Prioritization workshops, we refer to the zoonotic infections as diseases. This highlights that these priority zoonotic infections as capable of causing diseases; zoonotic infections unlikely to cause disease, were unlikely to be included as a priority zoonoses for the country. For consistency with the publications resulting from similar workshops, we would like to continue to use the term zoonotic diseases. 

6. Lines 176-177: There is a complete lack of explanation how the criteria were determined and very limited justification of them, despite the fact that this is described as a discussion. For instance, why was human mortality the final determinant for disease hazard/burden vs. DALY or something that also included agricultural impact? 

All workshop voting members were involved in the development of the criteria that were most important to their ministries and agencies/units for addressing zoonotic diseases in China. All voting members and workshop participants discussed and agreed on using these criteria to prioritize the top five zoonotic diseases in China. Following this discussion, workshop participants decided that the “Disease hazard/severity” criterion would solely focus on the human case fatality rate, which would help measure disease severity. They also decided that the “Economic impact” criterion would focus solely on animal burden in relation to production losses. Components for the other two criteria (“Epidemic scale and intensity” and “Prevention and control”) focused on both human and animal health. For the OHZDP process, not every criteria and question needs to incorporate components of each sector.

7. Why was vaccination the only control/prevention measure listed in the literature review and as the determining criteria? 

While vaccine was one of the key search terms used during the literature review, additional resources outside of the literature review for prevention and control measures for the zoonotic diseases were also available to the workshop participants. For the prevention and control criteria, workshop participants identified that this criterion should be measured by availability of human and animal vaccination and human treatment and medication. The focus was on prevention for humans and animals and then treatment for human if infected. 

8. What is the evidence behind internet hits (Baidu index) as a measure of “societal impact” particularly in a society that does not have free internet access to all sites and data? What are the inherent biases of these choices?

Workshop participants thought it was important to include “Social Impact” as a criterion for the zoonotic disease prioritization. To measure “Societal Impact”, workshop participants used the Baidu index as a proxy to assess public interest and social impact of the disease. The Baidu index is the most used online search engine in China and provides various automatically generated indexes. Most the of the population has access to the internet via mobile devices, particularly Smart phones. In 2018, approximately 73.02% of the population in China conducted an internet search using Baidu. These searches included health and disease specific search information. The Baidu search index is based on the search volume of users in Baidu, using keywords as the statistical objects, and the result represents the weighted sum of the search frequency of each keyword in Baidu web search. Diseases that received a lot of public attention (>700 ‘hit’ value) in the past year (1 year) were given the highest score of 4. Diseases that received little to no public attention (<100 ‘hit’ value) were given a score of 0. The index represents the weighted sum of the search frequency in a certain time period. Higher index values reflect a larger search volume, suggesting greater public interest and social impact. The one-year search timeframe could have influenced the score of this criterion. Certain diseases and their index score could be based on possible mentions from other news sources and/or international events as well as zoonotic diseases present during the year proceeding the workshop. Nevertheless, utilizing the Baidu index was a creative and unique way to measure societal interest and impact from these diseases. This approach can be used to monitor changes and emerging societal concerns over time. Please see lines 285-289.

9. Expand upon weights, and questions/answers within the text- how did the answers determined using the criteria in Table 3 lead to the raw scores in table 2? This must be explained so that these numbers make sense.

Thank you for your comment. In the methods section under “Disease weighting and final ranking”, a brief explanation of how the answers for each zoonotic disease and the criteria weighting are applied using decision tree analysis built into the OHZDP Tool. The full detailed methods for how the raw scores are attained are described in the original OHZDP publication (https://journals.plos.org/plosone/article?id=10.1371/journal.pone.0109986). To determine the ranked zoonotic disease list, we utilized a decision tree analysis built into the OHZDP Tool. Each criterion’s weight is multiplied by the score assigned for each disease and is weighted by the number of answer choices. The weighted scores for all 5 criteria for each disease are summed and normalized in relation to the zoonotic disease with the highest raw score. The OHZDP Tool then generated the ranked zoonotic disease list in order of the normalized final scores. These normalized final scores were calculated based on the inputs from the workshop participants. We have added in additional information on the methods into the revised manuscript in lines 152-156. 

10. Since the focus is on zoonotic diseases, it would be helpful to show the morbidity/mortality numbers for animals affected by the diseases chosen in addition to the human data.

Thank you for your comment. After much discussion, workshop participants agreed that the “Disease hazard/severity” criterion would solely focus on the human case fatality rate. Participants also decided that the “Economic impact” criterion would focus solely on animal burden in relation to production losses. Components of the other two criteria (“Epidemic scale and intensity” and “Prevention and control”) focused on both human and animal. For the OHZDP process not every criteria and question need to incorporate components of each sector. 

11. Minor points:

• Update the population of China stated in line 84.

China's population is still estimated to be 1.4 billion in 2019.

• Add international agencies that participated in the workshop among those listed in Table 1.

Thank you for this suggestion. All international agencies participating in the workshop have been added to Table 1 in revised manuscript.

• Why is Xinjiang Hemorrhagic Fever highlighted in Table 2?

Xinjiang Hemorrhagic Fever was not highlight in Table 2 (as following picture shows). There was no special reference for XHF during the workshop. 

 

Reviewer #2: 

1. I think it would be useful for the authors to demonstrate how the scaling system could be used to compare SARS-CoV2 to the other zoonotic diseases they evaluated. By my reckoning it comes out ahead of avian influenza, although the balance of the assessment would shift from early in the pandemic (no vaccine or therapies, little awareness) to later in the pandemic (vaccine and therapies and huge public interest). This would take a fairly dry assessment of an important exercise and turn it into a model for how this approach can help reprioritize public health actions.

Thank you for your suggestion. The process is flexible and adaptable to additional situations and can be applied to emerging infectious diseases as well. While the prioritization process focuses on endemic and emerging zoonotic diseases, countries are able to build and strengthen capacity for the priority zoonoses regardless if they are endemic or emerging. Having all relevant sectors work together using a One Health approach to develop a shared priority list as well as an action plan to address those priorities is helping to advance One Health, and this in turn helps to better prepare partners from multiple sectors to address an unknown pathogen or reemerging infectious diseases. We agree that a follow-up workshop – or analysis of COVID-19 as a priority zoonotic disease - should be conducted but recommend that this is developed as a separate project. We have provided additional comments in the discussion section of the revised manuscript to describe such a possible follow-up workshop. Please see lines 266-279.

2. Lines 47-48 “Disease severity” would be more understandable than “disease hazard” in describing this measure.

We appreciate this comment and have revised to “disease hazard” to “disease hazard/severity” in Lines 39 (as well as Table 3) in revised manuscript. We completely agree “disease severity” is more understandable to clarify the measure. Considering the language decided and agreed upon by the voting members during the workshop and the word using in workshop report, “hazard” is kept in this form.

3. Line 76 “resulting from spillover… which might be from an animal source.” The concept of spillover implies an animal source. Why is it necessary to qualify the statement?

Thank you for your comment. We have revised the statement to read: “The recent coronavirus disease 2019 (COVID-19) pandemic, resulting from the introduction of a novel coronavirus (SARS-CoV-2) to the human population, exemplifies challenges with emerging zoonotic diseases.”. Please see line 57-59 of revised manuscript. 

4. Line 109 As noted above, why not demonstrate how to apply the outcomes to the COVID-19 pandemic?

We appreciate this comment and agree that demonstrating the flexibility and adaptability of the tool is important. The main outcome of this initial workshop was to identify a list of priority zoonotic diseases that could be addressed using a One Health Approach. The list of 5 prioritized zoonotic diseases is a tangible outcome of the workshop. However, proposed work on surveillance, cross-sector training, and preparedness and response capacity should be applied to any new (or old) zoonotic disease. While the prioritization process focuses on endemic and emerging zoonotic diseases, countries are able to build and strengthen capacity for the priority zoonosis regardless of if they are endemic or emerging. Rather than generating a new score for COVID-19, we agree that the One Health approach could be applied to other emerging diseases not identified at the time of the workshop. We propose conducting follow-up workshop or separate project for this analysis. We have provided additional comments in the discussion section of the revised manuscript to describe such a possible follow-up workshop or analysis. Please see lines 194-196, 266-279.

5. Line 120 Note at the time of the meeting.

We add the date of meeting in Line 85 of revised manuscript.

6. What is the point of a 6-decimal place “normalized final score”? The ranking is based on categorical values from 0-4. The decimals imply a level of precision that are not reasonable distinctions. Particularly when the final choice was made based on “discussions among the voting members”.

Thank you for your comment. We have updated the content and Table 2 to reflect only 3-decimal place for the raw and normalized final scores. Please see line 172 and Table 2 in the revised manuscript.

7. Lines 194-197 This part of the discussion is a restatement of methods.

Thank you for this comment. We have deleted the these lines from the discussion section of the manuscript.

8. Lines 198-199 This is a restatement of results.

Thank you for this comment. We have deleted these lines from the discussion section of the manuscript. 

9. Lines 200-204 This appears to be a restatement of the premise behind the meeting.

Thank you for this comment. We have deleted these lines from the discussion section of the manuscript. 

10. Lines 315-317 - This should be part of an acknowledgement and not part of the conclusions.

Thank you for this comment. We moved these lines to the acknowledgement section in the revised manuscript.

11. In looking at the scoring system, it is not clear why the prevention and control measures are expressed as a negative relation, when all of the others are positive. I can understand that if a disease emerges for which there is no vaccine or therapy available (see COVID-19) then prioritizing their development is important. However, if (such as with rabies) vaccines and therapies are available, and the disease is not being effectively controlled, that demonstrates a critical need to better coordinate measures across One Health platforms. For the diseases that were ranked these are important considerations.

Thank you for your comment. Workshop participants decided it was most important to have zoonotic diseases without any prevention or control measure (vaccination or treatment/medicine) be ranked higher than zoonoses that have at least some prevention and control measures. However, after the priority zoonotic diseases were identified, participants also discussed One Health coordination and strengthening in China. During this discussion, participants recommended that there was a need to develop and implement multisectoral plans for disease control and prevention (including for outbreak control and response) for the priority zoonotic diseases. 

 

Reviewer #3: 

1.Methods: I would suggest modifying the Table 1 to include columns where they can check off if it is human, animal or environment, and possibly to indicate if they had a voting member. 

Thank you for your comment. We have updated the Table 1 to include each organization and corresponding role in the workshop. The table now includes voting members and advisors. Below is Table 1 that has included in the revised version of the manuscript. 

Table 1. Agencies participating in the One Health Zoonotic Disease Prioritization workshop, Beijing, China, May 15-17, 2019.

Voting Members

National Level

National Institute for Communicable Diseases Control and Prevention, China CDC

National Institute of Parasitic Diseases (NIPD), China CDC

Division of Infectious Disease, China CDC

China Animal Disease Control Centre

National Research Center for Wildlife Diseases

Eastern Region

Jiangsu Provincial CDC

Shandong Provincial CDC

Jiangsu Academy of Agricultural Sciences, Jiangsu Society of Animal Husbandry and Veterinary Medicine

Shandong Provincial Animal CDC

Northwestern Region

Qinghai Provincial CDC

Gansu Provincial CDC

Ningxia Hui Autonomous Region CDC

Qinghai Animal CDC

Lanzhou Veterinary Research Institute, Chinese Academy of Agricultural Sciences

Qinghai Academy of Animal Science and Veterinary Medicine

Southwestern Region

Yunnan Provincial CDC

Guangxi University of Chinese Medicine, School of Public Health and Management

Sichuan Provincial CDC

Guangxi Zhuang Autonomous Region Animal CDC

Yunnan Provincial Institute of Endemic Disease Prevention and Control

Sichuan Provincial Animal CDC

Facilitators 

US Centers for Disease Control and Prevention (Trainer)

National Institute for Communicable Diseases Control and Prevention, China CDC

Center for Global Public Health, China CDC

Jiangsu Provincial CDC

National Research Center for Wildlife Diseases

Yinchuan Animal CDC

Advisors and Other Participants

US Department of Health and Human Services China Office

US Department of Agriculture (USDA) China Office

US National Institutes of Health

US Centers for Disease Control and Prevention

Food and Agriculture Organization (FAO) China Office

WHO China Office

Center for Global Public Health, China CDC

National Institute for Communicable Diseases Control and Prevention, China CDC

1. The 9 groups were also unclear – how were they divided and why?

Thank you for your comment. The 28 voting members were divided into nine groups. Participating provinces were categorized into three regions. Each region (East, Northwest, and Southwest) had two voting groups (one for the human sector and one for the animal sector), making six voting groups. There were also three additional voting groups for the national level (one for the human health sector, one for the animal health sector, and one for the environmental sector). Please see lines 101 to 104, where we have provided additional detail in the establishing the nine voting groups.

3. Also the totals don’t seem to add up - there were 53 participants but then there were 28 voting members but 16 observers. This is 56. Just a bit more clarity here would be very helpful as to the relative roles and who was involved in the decision making.

Thank you for your comment. There were 53 participants at the workshop. This included 28 voting members, 16 advisors, and 9 facilitators. National and regional representatives involved in the organization of the workshop were identified as decision makers during the workshop. We have also updated Table 1 in the manuscript to better highlight the organizations participating in the workshop and their roles (voting members, advisors, facilitators). 

4. I would suggest that they include a bit more detail on the databases they searched, time frames searched, etc. While not necessarily a systematic review, it would be great to have as much detail reported about the search as possible. Also, were things like Promed databases used, etc.

Thank you for your comment. The literature review process incorporates information from the published and gray literature. Additional zoonotic disease information was obtained from informal reports, summary documents, or input from technical experts in country, and data provided by government and partner organizations, and other data sources such as ProMed, Global Incident Map, HealthMap, and news articles. 

We used the CNKI literature search engine database for our search of Chinese literature on the 30 important zoonotic diseases in China. We followed the same process that was used to search the English literature. For both the English and Chinese review, we focused on literature published within the last ten years. 

5. There are nine groups with 28 voting members – but the groups had to come to consensus so does this really mean there are only 9 votes?

Thank you for the comment. Yes, there were only 9 votes casted for the 28 voting members. The 28 voting members were divided into nine groups. Participating provinces were categorized into three regions. Each region (East, Northwest, and Southwest) had two voting groups (one for the human sector and one for the animal sector), making six voting groups. There were also three additional voting groups for the national level (one for the human health sector, one for the animal health sector, and one for the environmental sector).We have added this information to the Methods section of the of revised manuscript. Please see lines 101-104.

6. This is a very nice discussion and systematic method of prioritizing current identified threats. However, as was pointed out SARS-CoV-2 and coronaviruses were not included at the time of the workshop but emerged as a significant threat. Some discussion around how to increase the capacity and response for unknown threats would also be useful.

Thank you for your comment. While the prioritization process focuses on endemic and emerging zoonotic diseases, countries are able to build and strengthen capacity for the priority zoonoses regardless of whether they are endemic or emerging. Having all relevant sectors work together using a One Health approach to develop a shared priority list as well as an action plan to address those priorities is helping to advance One Health, and this in turn helps to better prepare partners from multiple sectors to address an unknown pathogen or reemerging infectious diseases. We agree that a follow-up workshop – or analysis of COVID-19 as a zoonotic disease - should be conducted but recommend that this developed as a separate project. We have provided additional comments in the discussion section of the revised manuscript to describe such a possible follow-up workshop. Please see lines 266-279.

While the prioritization process focuses on endemic and emerging zoonotic diseases, when countries are able to build and strengthen capacity for the priority zoonoses regardless of if they are endemic or emerging, countries are building better capacity to address whatever the next new pathogen is. Having all relevant sectors work together using a One Health approach to develop a shared priority list as well as an action plan to address those priorities is helping to advance One Health, and this in turn helps to better prepare partners from multiple sectors to address an unknown pathogens or reemerging infectious diseases. Some comments were added in the discussion of Line 195-196 of revised manuscript.

---

## [Decision Letter · Decision Letter 1]

3 Aug 2021

PONE-D-20-41070R1

Using a One Health Approach to Prioritize Zoonotic Diseases in China, 2019

PLOS ONE

Dear Dr. Jing,

Thank you for submitting your manuscript to PLOS ONE. After careful consideration, we feel that it has merit but does not fully meet PLOS ONE’s publication criteria as it currently stands. Therefore, we invite you to submit a revised version of the manuscript that addresses the points raised during the review process.

The reviewer recommends that you make minor revisions to your manuscript. Some of the concerns relate to comments that were raised in the previous review while others are new. Please attend to all of them adequately and resubmit the revised manuscript as advised in this letter.

We look forward to receiving your revised manuscript.

Kind regards,

Martin Chtolongo Simuunza, PhD

Academic Editor

PLOS ONE

Journal Requirements:

Reviewers' comments:

Reviewer's Responses to Questions

**Comments to the Author**

1. If the authors have adequately addressed your comments raised in a previous round of review and you feel that this manuscript is now acceptable for publication, you may indicate that here to bypass the “Comments to the Author” section, enter your conflict of interest statement in the “Confidential to Editor” section, and submit your "Accept" recommendation.

Reviewer #1: (No Response)

Reviewer #2: All comments have been addressed

2. Is the manuscript technically sound, and do the data support the conclusions?

Reviewer #1: Yes

Reviewer #2: Yes

3. Has the statistical analysis been performed appropriately and rigorously? 

Reviewer #1: N/A

Reviewer #2: Yes

4. Have the authors made all data underlying the findings in their manuscript fully available?

Reviewer #1: Yes

Reviewer #2: Yes

5. Is the manuscript presented in an intelligible fashion and written in standard English?

Reviewer #1: Yes

Reviewer #2: Yes

6. Review Comments to the Author

Reviewer #1: The authors addressed most of our comments. Explanations were added for the categories and inclusions of workshop participants as well as the formation of the initial zoonotic disease list. Statements were added for expanding upon weighting and ranking methods. These additions improved the methods section.

An explanation was added concerning COVID-19, the timing of the 2019 workshop, and considerations for the future. This addition helps readers understand the lower ranking of SARS.

Minor edits:

An explanation of the Baidu index was added, although more language regarding the bias of this approach would be useful to readers.

The authors’ reply to comment 1 could be incorporated into the manuscript text regarding participation of certain provinces and geographic locations. Including an explanation about provincial participation and potential biases would strengthen the methods section.

Table 1: The number of agencies is less than the number of participants. Could you add the number of participants per agency?

The previous comment concerning terminology of diseases versus infections was only partially addressed. The authors replied that the terminology was used for consistency with previous OHZDP workshops. Changing the terminology to correctly identify infections or diseases should be implemented. At the very least, a statement regarding this ‘consistency’ of terminology should be included.

In the discussion, limitations or implications of use of the chosen criteria (not just social impact) should be added.

Reviewer #2: (No Response)

7. PLOS authors have the option to publish the peer review history of their article (what does this mean?). If published, this will include your full peer review and any attached files.

Reviewer #1: No

Reviewer #2: No

---

## [Author Response · Author response to Decision Letter 1]

25 Sep 2021

Response to Journal and Reviewers

Journal Requirements

Response to Journal

Thanks for providing the requirements. Accordingly, the reference list has been reviewed and revised as follows:

1.Reference [3] is revised to include complete information on line 346-347.

2.Reference [30] is provided with updated URL on line 417, to replace formerly invalid URL.

3.All URL of the reference has been checked for validity, and ‘Last accessed MMMDD,YYYY’ was updated for references [3][10][13][18][25][28][29][30] . 

4.No reference has been found retracted, by searching references in https://pubmed.ncbi.nlm.nih.gov/.

Reviewer 1 

The authors addressed most of our comments. Explanations were added for the categories and inclusions of workshop participants as well as the formation of the initial zoonotic disease list. Statements were added for expanding upon weighting and ranking methods. These additions improved the methods section. An explanation was added concerning COVID-19, the timing of the 2019 workshop, and considerations for the future. This addition helps readers understand the lower ranking of SARS.

Response to Reviewer 1 

Thanks for reviewer’s comment. We deeply appreciate your review on the paper. We have studied the comments carefully and revised the manuscript that takes all the points into account. The following addresses each of the comments item by item. We would like to express our appreciation for your professional comments during each revision, which have really helped the paper improved much. 

Minor edits

Q1: 

An explanation of the Baidu index was added, although more language regarding the bias of this approach would be useful to readers.

The authors’ reply to comment 1 could be incorporated into the manuscript text regarding participation of certain provinces and geographic locations. Including an explanation about provincial participation and potential biases would strengthen the methods section.

A1:

Thanks for your comments. Certain provinces were added to geographic locations on line 100-101,

“The eight provinces represent the primary ecological and socio-demographic regions in the east (Jiangsu and Shandong), northwest (Qinghai, Ningxia, and Gansu ) and southwest (Guangxi, Sichuan and Yunnan) of China respectively. ”

Thanks for your suggestion of incorporating our reply to previous comment 1 into this revision about provincial participation and potential biases . As the example of our previous reply, though staffs from Heilongjiang and Hunan Province were unnable to attend the workshop, Yin Wenwu (who attended) and his team from China CDC worked closely with these provinces, the high-risk province for rabies and anthrax respectively. 

Accordingly, it is revised on line 101-103: 

“Not all provinces were able to participate. However, national managers participating in the workshop were able to provide input on behalf of the human, animal, and environmental health sector staff in the provinces unable to attend. ” 

Q2:

Table 1: The number of agencies is less than the number of participants. Could you add the number of participants per agency?

A2:

Thanks for your comments. Accordingly to the comments, the number of participants has been added to each agency in table 1.

Voting Members (No. participants)

National Level 

National Institute for Communicable Diseases Control and Prevention, China CDC (2)

National Institute of Parasitic Diseases (NIPD), China CDC (1)

Division of Infectious Disease, China CDC (1)

China Animal Disease Control Centre (1)

National Research Center for Wildlife Diseases (1)

Eastern Region

Jiangsu Provincial CDC (2)

Shandong Provincial CDC (1)

Jiangsu Academy of Agricultural Sciences, Jiangsu Society of Animal Husbandry and Veterinary Medicine (1)

Shandong Provincial Animal CDC (1)

Northwestern Region

Qinghai Provincial CDC (3)

Gansu Provincial CDC (1)

Ningxia Hui Autonomous Region CDC (1)

Qinghai Animal CDC (2)

Lanzhou Veterinary Research Institute, Chinese Academy of Agricultural Sciences (1)

Qinghai Academy of Animal Science and Veterinary Medicine (3)

Southwestern Region

Yunnan Provincial CDC (1)

Guangxi University of Chinese Medicine, School of Public Health and Management (1)

Sichuan Provincial CDC (1)

Guangxi Zhuang Autonomous Region Animal CDC (1)

Yunnan Provincial Institute of Endemic Disease Prevention and Control (1)

Sichuan Provincial Animal CDC (1)

Q3: 

The previous comment concerning terminology of diseases versus infections was only partially addressed. The authors replied that the terminology was used for consistency with previous OHZDP workshops. Changing the terminology to correctly identify infections or diseases should be implemented. At the very least, a statement regarding this ‘consistency’ of terminology should be included.

A3:

Your understanding for the consistency with previous OHZDP workshops is much appreciated. A statement regarding this ‘consistency’ is included in method section on line 129-131 as,

“For consistency with publications resulting from previous workshops, we use the term zoonotic diseases rather than zoonotic infections. The term highlights that these priority zoonotic infections are capable of causing disease.”

Q4:

In the discussion, limitations or implications of use of the chosen criteria (not just social impact) should be added.

A4:

Thanks for reviewer’s comments, the limitations of using chosen criteria is important and should be added. The method of One Health Zoonotic Disease Prioritization has been utilized in more than 25 locations, including USA, Uganda, Kenya, Kenya, etc [13-17]. Much experiences have been accumulated by the workshop and are inherited by the workshop advisor. The national level workshop did not aim to address all possible zoonotic disease prioritization criteria since the OHZDP methods focus on the top five criteria of greatest concern.Participants discussed many different criteria during the workshop and agreed on the five criteria most likely associated the potential zoonotic disease threat in the country.

Each criteria has a corresponding question, which is required an ordinal or multinomial answer to measure the diseases. When deciding criteria, the participants bear in mind to cover as much diseases as possible, but still a few disease has no answer to certain criteria’s question. Taking criteria “Economic impact” as example, livestock production loss is the question used for measurement. If livestock production loss is not available, case fatality rate and mortality were used as proxies; if China-specific data is not available, global data is used instead (see line 454-456, footnote * of Table 3). Future workshops – including those conducted at the regional or provincial level - could develop and apply more targeted and specific criteria for identifying a local list of priority zoonotic diseases. 

According revision is added on line 304-310.

---

## [Decision Letter · Decision Letter 2]

26 Oct 2021

Using a One Health Approach to Prioritize Zoonotic Diseases in China, 2019

PONE-D-20-41070R2

Dear Dr. Jing,

We’re pleased to inform you that your manuscript has been judged scientifically suitable for publication and will be formally accepted for publication once it meets all outstanding technical requirements.

Kind regards,

Martin Chtolongo Simuunza, PhD

Academic Editor

PLOS ONE

Additional Editor Comments (optional):

Reviewers' comments:

Reviewer's Responses to Questions

**Comments to the Author**

1. If the authors have adequately addressed your comments raised in a previous round of review and you feel that this manuscript is now acceptable for publication, you may indicate that here to bypass the “Comments to the Author” section, enter your conflict of interest statement in the “Confidential to Editor” section, and submit your "Accept" recommendation.

Reviewer #1: All comments have been addressed

Reviewer #2: All comments have been addressed

2. Is the manuscript technically sound, and do the data support the conclusions?

Reviewer #1: Yes

Reviewer #2: (No Response)

3. Has the statistical analysis been performed appropriately and rigorously? 

Reviewer #1: Yes

Reviewer #2: (No Response)

4. Have the authors made all data underlying the findings in their manuscript fully available?

Reviewer #1: Yes

Reviewer #2: (No Response)

5. Is the manuscript presented in an intelligible fashion and written in standard English?

Reviewer #1: Yes

Reviewer #2: (No Response)

6. Review Comments to the Author

Reviewer #1: (No Response)

Reviewer #2: (No Response)

7. PLOS authors have the option to publish the peer review history of their article (what does this mean?). If published, this will include your full peer review and any attached files.

Reviewer #1: No

Reviewer #2: No

---

## [Editor Report · Acceptance letter]

11 Nov 2021

PONE-D-20-41070R2 

Using a One Health Approach to Prioritize Zoonotic Diseases in China, 2019 

Dear Dr. Jing:

I'm pleased to inform you that your manuscript has been deemed suitable for publication in PLOS ONE. Congratulations! Your manuscript is now with our production department. 

Kind regards, 

on behalf of

Dr. Martin Chtolongo Simuunza 

Academic Editor

PLOS ONE